# An Insight into the Mechanism of Holamine- and Funtumine-Induced Cell Death in Cancer Cells

**DOI:** 10.3390/molecules25235716

**Published:** 2020-12-03

**Authors:** Jelili A. Badmus, Okobi E. Ekpo, Jyoti R. Sharma, Nicole Remaliah S. Sibuyi, Mervin Meyer, Ahmed A. Hussein, Donavon C. Hiss

**Affiliations:** 1Department of Medical Biosciences, University of the Western Cape, 7535 Bellville, Western Cape, South Africa; jabadmus@lautech.edu.ng (J.A.B.); oekpo@uwc.ac.za (O.E.E.); 2DSI/Mintek-Nanotechnology Innovation Centre-BioLabels Node, Department of Biotechnology, University of the Western Cape, 7535 Bellville, Western Cape, South Africa; jrsharma@uwc.ac.za (J.R.S.); nsibuyi@uwc.ac.za (N.R.S.S.); memeyer@uwc.ac.za (M.M.); 3Department of Chemistry, Cape Peninsula University of Technology, 7535 Bellville, Western Cape, South Africa; mohammedam@cput.ac.za

**Keywords:** Apocynaceae, anticancer, apoptosis, holamine, funtumine, steroidal alkaloids

## Abstract

Holamine and funtumine, steroidal alkaloids with strong and diverse pharmacological activities are commonly found in the Apocynaceae family of *Holarrhena*. The selective anti-proliferative and cell cycle arrest effects of holamine and funtumine on cancer cells have been previously reported. The present study evaluated the anti-proliferative mechanism of action of these two steroidal alkaloids on cancer cell lines (HT-29, MCF-7 and HeLa) by exploring the mitochondrial depolarization effects, reactive oxygen species (ROS) induction, apoptosis, F-actin perturbation, and inhibition of topoisomerase-I. The apoptosis-inducing effects of the compounds were studied by flow cytometry using the APOPercentage^TM^ dye and Caspase-3/7 Glo assay kit. The two compounds showed a significantly greater cytotoxicity in cancer cells compared to non-cancer (normal) fibroblasts. The observed antiproliferative effects of the two alkaloids presumably are facilitated through the stimulation of apoptosis. The apoptotic effect was elicited through the modulation of mitochondrial function, elevated ROS production, and caspase-3/7 activation. Both compounds also induced F-actin disorganization and inhibited topoisomerase-I activity. Although holamine and funtumine appear to have translational potential for the development of novel anticancer agents, further mechanistic and molecular studies are recommended to fully understand their anticancer effects.

## 1. Introduction

Plants are almost certainly the oldest source of agents exploited by humans to combat several ailments and improve vigour and vitality. Considerable scientific evidence supports the notion that secondary metabolites of plants such as alkaloids are responsible for their observed therapeutic effects [1]. Alkaloids are widely distributed in higher plants belonging to the *Apocynaceae*, *Leguminosae*, *Papaveraceae*, *Ranunculaceae* and *Lamiaceae* families [2,3]. Alkaloids are structurally diverse compounds with a nitrogen atom coordinated in a heterocyclic ring. They are reputed for their ability to interact with a wide assortment of biomolecules, because their building blocks are made up of different amino acids and alkyl radicals replaced by hydrogen atoms [4]. The pronounced pharmacological activity of alkaloids is due to their structural diversity and seemingly indiscriminate interaction with a variety of molecules to modulate stability and reduce toxicity. The medicinal importance of alkaloids continues to gain prominence due to their wide spectrum of physiological properties [5]. Several medicinal attributes ascribed to alkaloids such as camptothecin and vinblastine have contributed to their development into therapeutic agents against cancer [6]. Steroidal alkaloids constitute an important class of compounds isolated as glycoalkaloids in higher plant families as well as some amphibians and marine invertebrates [7]. Several mechanisms of action of steroidal alkaloids associated with their anticancer, anti-inflammatory, antimicrobial, antithrombotic, antiandrogenic, and antiarrhythmic activities have been documented [8]. Steroidal alkaloids obtained from plants belonging to the Apocynaceae (dogbanes) family are known for their strong biological effects, including cancer cell killing [8]. Some of the known mechanisms of action of steroidal alkaloids include the initiation of apoptosis (through the induction of Bax expression, Bcl-2 suppression, and PARP-1 initiation), cell cycle arrest (at G0/G1 and G2/M check points), inhibition of cell-signaling protein (MMP-2/9 and AKT), and activation of transcriptional factors (p21WAF1/CIP1 and checkpoint kinase-2) [9]. Holamine and funtumine are pregnene-type steroidal alkaloids commonly found in the Apocynaceae family of *Holarrhena* with strong and diverse pharmacological activities [10,11,12]. These alkaloids have demonstrated a significant selective cytotoxicity against human cancer cells (HeLa, MCF-7 and HT-29) when compared with non-cancerous cells (i.e., normal KMST-6 fibroblasts). Both compounds were found to reduce cancer cell viability to 50% at concentrations ranging from 22 to 53 µM [11]. Their cytotoxicity in cancer cell lines was associated with cell cycle arrest at the G0/G1 and G2/M phases [11]. Compounds from some medicinal plants are known to induce cytotoxicity through the induction of apoptotic pathways triggered by mitochondrial dysfunction, the increased production of reactive oxygen species (ROS), and the inhibition of topoisomerase-I activity in tumor cells [13,14]. This study sought to investigate the effects of holamine and funtumine on apoptosis, oxidative stress, mitochondrial depolarization, actin filament disorganization, and topoisomerase-I inhibition in selected cancer cells
.

## 2. Results and Discussion

We have previously shown, using the MTT and bromodeoxyuridine (BrdU) incorporation assays, that holamine and funtumine (Figure 1) inhibit the growth of the MCF-7, HeLa, HT-29, and KMST-6 cell lines [11]. This study also showed that both compounds block cell cycle progression in cancer cells, thus demonstrating promising anticancer properties [11]. This study further explored the effects of holamine and funtumine on cancer cells by evaluating the induction of apoptosis, the activation of caspase-3/7, ROS production (oxidative stress induction), the perturbation of mitochondrial functions, actin filament disorganization, and the inhibition of topoisomerase-I. The cytotoxicities of these alkaloids in cancer and non-cancer cells were further confirmed using the MultiTox-Glo multiplex cytotoxicity assay, reputed to have a higher sensitivity and precision than the MTT assay. The Multiplex assay sequentially measures the protease activities of either viable or membrane-compromised cells using the cell-permeant fluorogenic substrate, glycyl-phenylalanyl-aminofluorocoumarin, and the luminogenic substrate, alanyl-alanyl-phenylalanyl-aminoluciferin, respectively. Figure 2 shows that the two steroidal alkaloids induced a significant cytotoxic effect in the cancer cells in the range of 15–30 µg/mL and in the non-cancer cells at the highest concentration (30 µg/mL). Both compounds significantly induced cytotoxicity at the highest dose of 30 µg/mL, while different levels of cytotoxicity were observed at 15 µg/mL. Holamine at 15 µg/mL induced significant cytotoxicity in HT-29 (cytotoxicity = 60.69%; viability = 39.31%), MCF-7 (cytotoxicity = 53.40%; viability = 46.59%), HeLa (cytotoxicity = 51.39%; viability = 48.61%), and KMST-6 (cytotoxicity = 20.06%; viability = 79.94%). The cytotoxicity induced by funtumine at the same concentration was lower in HT-29 (cytotoxicity = 53.87%; viability = 46.3%), MCF-7 (cytotoxicity = 44.41%; viability = 55.59), HeLa (cytotoxicity = 36.63%; viability = 63.37%), and KMST-6 (cytotoxicity = 24.00%; viability = 76%). Holamine at 15 µg/mL induced a cytotoxicity of more than 50% in all the cancer cell lines, except for KMST-6, while the same effect was only observed with MCF-7 when treated with funtumine. Both compounds induced cytotoxicity in the hierarchy of HT-29 > MCF-7 > HeLa > KMST-6. These results appear to confirm our earlier observations of the selective cytotoxicity of the compounds to cancer cells relative to non-cancer cells [11]. A high selectivity index towards cancer cells is an important consideration when assessing the medicinal and therapeutic attributes of bioactive moieties from natural sources. Compounds with high selectivity indices impart lower toxicities to the sensitive and regulated normal proliferative cell compartments in the tissue microenvironment and afford a cost-effective way to rationalize a minimum level of tumor resistance and side effects in response to chemotherapy [15].

Mitochondria play diverse and crucial roles in cell survival and death and are also critical for oncogenesis and the sustained progression of malignant cells. Compounds that target mitochondria are important for the ablation of malignant cells because of their ability to induce excessive reactive oxygen species (ROS), leading to oxidative stress and eventual mitochondrial damage [16]. Cancer cells strive in environments with increased ROS and the concomitant elevation of antioxidant proteins to maintain the requisite oxidation-reduction (REDOX) balance. Cancer cells target drug actions either by disrupting the activity of antioxidant proteins or inducing ROS generation to disturb the REDOX balance, a potential for the application of the selective cytotoxicity index principle [17]. The probable influence of holamine and funtumine on mitochondrial functions in relation to ATP generation was assessed using the MitoToxGlo™ assay. This assay measures biomarkers linked to cell membrane integrity alteration and cellular ATP levels due to drug treatments and predicts the structural-functional status of mitochondria. It also evaluates the membrane integrity of treated cells by quantifying the protease activity associated with necrosis (Figure 3). The number of necrotic cells was negligible in all the cell lines, except for the HT-29 cells treated with the highest concentration of funtumine. The compounds did not induce necrotic effect in any of the cell lines tested, while the moderate cytotoxicity observed in the MCF-7 cell line was not statistically significant. However, funtumine significantly reduced ATP levels as a function of mitochondrial modulation in HeLa and HT-29 cells, while holamine only affected ATP levels in HeLa cells at the highest concentration (30 µg/mL). Thus, the cytotoxicity of these compounds did not appear to be related to necrosis, as no changes in membrane integrity were observed. This implies that the cytotoxic effects of these compounds might be associated with mechanisms other than necrosis. The reduction in the ATP levels in HT-29 and HeLa cells suggests decreased mitochondrial function induced by both compounds. However, the implications of these observations are not generally applicable, as the compounds did not induce the same effect in MCF-7 as in the other two cell lines. Furthermore, the compounds caused a reduction in mitochondrial activity of cells after a 12-h treatment, as shown by the reduced intensity of the Mitotracker red stain in Figure 4. The photomicrograph show that the low imparted intensity of the mitochondrial stain in HeLa cells treated with the two compounds was more reduced when compared with the untreated control. This suggests that the effects of the two compounds could be related to the modulation of mitochondrial functions, particularly mitochondrial respiration. The importance of mitochondria in cell survival and death has made it an attractive target organelle for the development of anticancer agents to restore normal cell physiology and attenuate the progression to malignancy [18]. Drugs with the potential to perturb mitochondrial functions might represent an important targeted cancer chemotherapeutic approach [19]. Such drugs function by depleting the ATP levels, causing the dephosphorylation of pro-apoptotic proteins, the mobilization of Bax to the mitochondria, the permeabilization of the outer mitochondrial membrane, and eventual cell death [20].

These results further imply that holamine and funtumine might inhibit one or more processes that promote unregulated proliferation of cancer cells and can therefore be considered as good lead compounds for cancer treatment in view of their pronounced cytotoxic activities against the panel of cancer cells. Cancer treatment requires therapeutic agents with an efficient and selective toxicity towards cancer cell lines, hence plants are attracting intense interest for the development of new classes of anticancer agents [21]. The cytotoxic spectrum of holamine and funtumine might affect cellular processes associated with either cancer cell survival or resistance to cell death. Cancer cells evade cell death by some of the following processes: induced external growth factors, intracellular matrix signalling via integrin and Ras protein, and mutation-derived constitutive mitogenic signals leading to growing neoplasms that cause the destruction and atrophy of the surrounding normal tissue and adjacent organs [22,23,24,25,26]. Thus, any compound that can disrupt these processes may have successful applications in the treatment and eradication of cancer. Resistance to apoptosis is one of the mechanisms acquired by cancer cells and is thus an indisputable target for the induction of cell death and subsequently also for therapeutic intervention [27,28,29,30]. The molecular mechanisms of the two steroidal alkaloids remain elusive, as Figure 3 and Figure 4 demonstrate that funtumine and holamine induce mitotoxicity in the cancer cells studied here and, as such, their effects on apoptotic biomarkers warrant further investigation. Cells undergoing apoptosis are characterized by various cellular markers. In the current study, phosphatidylserine (PS) externalization and increase in caspase activity were analyzed using the APOPercentage^TM^ dye and Caspase-Glo^®^-3/7 assays, respectively. The APOPercentage™ (disodium salt of 3,4,5,6,-tetrachloro-2,4,5,7-tetraiodofluorescein) dye can only be taken up by apoptotic cells at the late stage during PS externalization [31]. As shown in Figure 5, the two compounds induced cell death through apoptosis. The induction of apoptotic cells was significantly higher in cancer cells compared with in non-cancer cells and occurred in a dose-dependent manner. However, the results presented in Figure 5 show that cancer cells react to the apoptotic influence of these two compounds differently. This scenario is not completely unexpected, because they expressed different sensitivities towards the cytotoxic effects of these two compounds as presented here and as reported previously for another compound [32]. All the three cancer cell lines showed significant apoptotic sensitivity to holamine at 15 µg/mL, while MCF-7 appeared more sensitive to funtumine-induced apoptotic effects at 3.75 and 7.5 µg/mL concentrations.

Caspase-3/7 activation was assessed to validate the apoptotic effects of holamine and funtumine on cancer cells using the Caspase-3/7 Glo^TM^ assay kit. The activation of executioner caspases are considered as one of the hallmarks of apoptosis [33]. The intrinsic and extrinsic apoptotic pathways converge at the point where the executioner caspase, caspase-3 is activated [34]. Therefore, caspase-3 is an important molecular biomarker for the assessment of apoptosis [35]. In this study, caspase-3/7 activity following 12- and 24-h exposure to the compounds was considerably higher in HeLa cells at higher concentrations (Figure 6). The caspase activity in the HT-29 and MCF-7 cells was not significant with all the treatments. The induction of caspase-3/7 in the panel of cancer cells by both compounds can be related to their cytotoxic effects. The lower cytotoxic impact of both compounds on HeLa cells compared to other cell lines shows that, at the time of the cytotoxic evaluation, the cells were still in the early stage of apoptosis. The fold increase in caspase-3/7 activity in HeLa cells after 12 h is higher at 30 µg/mL, while after 24 h 15 µg/mL showed a higher fold increase. This implies that caspase-3/7 can be detected when the cells are not at the late stage of apoptosis. The activation of caspase-3/7 thus confirms that the compounds induced cell death through the apoptosis pathway. Apoptosis plays an important role in normal cell survival and in the maintenance of development and homeostasis [27,36,37,38], and is regarded as an important process to obliterate cancer cells [29,39]. Agents with apoptosis-inducing ability have been used in cancer therapy, as apoptosis pathways are frequently impaired in many cancers [27,30].

Reactive oxygen species (ROS) modulate the delicate cellular REDOX balance. Due to the chemical reactivity of ROS, its excessive production can lead to oxidative stress. They destabilize mitochondria and attack various cellular components including DNA, leading to the generation of oxidized and modified cellular components, and the eventual induction of apoptosis [40,41]. Increased ROS production is one of the mechanisms of some forms of conventional treatments—such as radiation, etoposide, bleomycin, and anthracyclines—that stimulate ROS production and the subsequent inhibition of cancer cell proliferation [40]. The production of ROS within the cells was measured using the CM-H_2_DCFDA fluorogenic probe which permeates freely into cells and when its acetate groups are cleaved by intracellular esterases to a highly fluorescent compound 2,7-dichlorofluorescin (DCF), which can be detected and quantified by flow cytometry or fluorescence microscopy. The two compounds induced significant fold increases in the mean fluorescence intensity (MFI) of the ROS probe at 12- and 24-h treatments in all the cells. However, ROS productions in HeLa and HT-29 cells treated with the two compounds were significantly higher (Figure 7). The production of ROS was confirmed by fluorescence microscopy (Figure 8). These results raise the possibility that the induction of apoptosis in the cancer cells might be due to the increased ROS and subsequent oxidative stress caused by the two compounds. The induction of ROS has been reported for some alkaloids with oxidizing properties [42]. Cathachuine and rohitukine alkaloids isolated from *Catharanthus roseus* (L.) G. Don and *Dysoxylum binectariferum* (Hook f.), respectively, have been shown to induce apoptosis in cancer cells through intracellular ROS generation [1].

Funtumine induced mitotoxicity without necrosis in the HeLa and MCF-7 cells, while primary necrosis was found in the HT-29 cells. The sensitivity of the different cell lines to the mitochondria-induced toxicity of the compounds might be related to the differences in ROS generation and mitochondrial dysfunction in the cells which make them more susceptible to oxidative stress triggered by the compounds [43]. Consistent with the results of ROS production in the treated cells, as presented in Figure 7 and Figure 8, it was observed that both the HeLa and HT-29 cells exhibited a significant increase in ROS induced by the treatment with the two alkaloids.

The treatment of cancer cells with both holamine and funtumine also caused disruptions in the F-actin filaments in HeLa cells compared with the well-organized stress fibres of actin filaments in the untreated controls (Figure 9). It is known that, during the cell blebbing phase, actin and myosin filaments slide over each other, leading to the contraction of the cell border toward the center [44], as observed in the treated cells.

This implies that both compounds can disrupt the actin filament organization, which could lead to cell death through apoptosis. Drugs that induce apoptosis and DNA breakage are known to cause nuclear alteration by the disruption of the cytoskeletal organization [44,45]. The inhibition of DNA topoisomerase-I leads to cell cycle arrest and cell death by apoptosis due to its role in relaxing DNA supercoiling during processes such as cell replication, recombination, transcription, chromatin assembly, and chromosome partitioning at critical cell-cycle events [42,46,47]. The present work, however, shows that holamine and funtumine inhibited DNA topoisomerase-I, as presented in Figure 10. This result suggests that the cytotoxic and apoptotic activities of these compounds might be related to topoisomerase-I inhibition. DNA topoisomerase-I and -II are excellent targets of clinically significant classes of anticancer drugs and effective strategies for cancer therapy [48]. Camptothecin and its derivatives, topotecan and irinotecan, are known topoisomerase inhibitors through which such anticancer activities are induced. Some alkaloids—e.g., pallidine and scoulerine isolated from *Corydalis saxicola* and epibeberine, and gloenlandcine extracted from *Coptis* rhizomes—have also been found to inhibit DNA topoisomerase-I [49,50].

## 3. Materials and Methods

### 3.1. Chemicals and Reagents

The APOPercentage™ kit was acquired from Biocolor Ltd. (Carrickfergus, Ireland); the Caspase-Glo^®^-3/7 assay kit, MitoToxGlo^™^, and MultiTox-Glo multiplex kit were acquired from Promega (Madison, WI, USA); the Topoisomerase-I relaxation assay kit was acquired from TopoGen (Buena Vista, CO, USA); the Annexin V-FITC apoptosis kit was acquired from BD Pharmingen (San Diego, CA, USA); the 5- (and 6) -chloromethyl-2′,7′-dichlorofluorescein diacetate acetyl ester (CM-H_2_DCFDA) and MitoTracker Red (CMXRos) were acquired from Molecular Probes (Invitrogen, OR, USA); the DAPI (4′,6-diamidino-2-phenylindole), TRITC-conjugated phalloidin, and anti-vinculin were acquired from Merck Millipore (Darmstadt, Germany); Hank’s balanced salt solution (HBSS), Dulbecco’s Modified Eagle’s Medium (DMEM), and phosphate buffered saline (PBS) were acquired from Gibco (USA); penicillin/streptomycin, Triton X-100, Tween-20, and paraformaldehyde were acquired from Sigma-Aldrich (St. Louis, MO, USA); and NucBlue Live ReadyProbes^TM^ Reagent was purchased from Thermo Fischer (Johannesburg, South Africa). All other chemicals and reagents used in this study were of analytical grade. 

### 3.2. Isolation of Holamine and Funtumine

The isolation of holamine and funtumine from the leaves of *Holarrhena floribunda* has been described previously [11].

### 3.3. Cell Culture Maintenance

Human cancer cell lines, viz., HT-29 (colon adenocarcinoma), HeLa (cervical cancer), MCF-7 (breast adenocarcinoma), and human non-cancer skin fibroblast (KMST-6) cells were a kind gift from Prof Denver Hendricks (Department of Clinical and Laboratory Medicine, University of Cape Town, South Africa). The cell lines were maintained in DMEM supplemented with 10% foetal bovine serum (FBS) and 1% penicillin/streptomycin (100 U/mL of penicillin and 100 µg/mL of streptomycin) and cultured as monolayers at 37 °C in 80% relative humidity and a 5% CO_2_ atmosphere.

### 3.4. MultiTox-Glo™ Assay

The cytotoxicity of the compounds against the four cell lines was assessed using the MultiTox-Glo multiplex cytotoxicity assay (Promega, Madison, WI, USA) according to the manufacturer’s instructions. Briefly, the cells (HT-29, HeLa, MCF-7, and KMST-6) were seeded in 96-well white-walled cell culture plates at a density of 5 × 10^4^ cells/mL (100 µL cell suspension per well) and treated for 24 h with increasing concentrations (0–30 µg/mL) of holamine or funtumine. The cytotoxicity of the compounds was assessed after 15 min of treatment using the MultiTox-Glo multiplex cytotoxicity reagent. The percentage live and dead cells were quantified by measuring the luminescence on a PolarStar Omega Plate Reader (BMG Labtech, Offenburg, Germany).

### 3.5. APOPercentage^TM^ Assay

The APOPercentage™ apoptosis assay (Biocolor Ltd., Carrickfergus, Ireland) was used to quantify the apoptosis in cell cultures treated with holamine or funtumine, following a protocol described by Meyer et al. [31]. The cells were seeded in 12-well cell culture plates at a density 1 × 10^5^ cells/mL and were treated with 10 to 30 µg/mL of the compounds for 24 h. After treatment, the apoptotic cells were analyzed using a Becton Dickinson FACScan instrument (BD Biosciences Pharmingen, San Diego, CA, USA). A minimum of 10,000 events per sample were recorded. The data were analyzed using the CELLQuest PRO^®^ software (BD Biosciences Pharmingen, San Diego, CA, USA).

### 3.6. Caspase Glo^®^-3/7 Assay

The Caspase-Glo^®^-3/7 assay kit (Promega, Madison, WI, USA) was used to assess the caspase-3 and -7 activity in cells treated with holamine or funtumine. The assay was performed as described by the manufacturer. Briefly, 5 × 10^4^ cells/mL (100 µL cell suspension per well) were seeded in a white-walled 96-well microplate and incubated for 12 and 24 h, respectively. After treatment with increasing concentrations (0–30 µg/mL) of the two compounds, an equal volume of Caspase-Glo^®^-3/7 reagent was added to each well, and the luminescence signal was recorded using a GloMax Multi Detection System (Promega, Madison, WI, USA) after 1 h of incubation.

### 3.7. Evaluation of ROS

The intracellular ROS production was evaluated using the fluorogenic molecular probe 5-(and 6)-chloromethyl-2′,7′-dichlorofluorescein diacetate acetyl ester (CM-H_2_DCFDA) and analyzed by flow cytometry and fluorescence microscopy.

#### 3.7.1. Flow Cytometric Analysis

The cells were cultured in a 6-well plate at a density of 2 × 10^5^ cells/mL. The cells were treated with 15 µg/mL of the compounds for 12 and 24 h, respectively. After the treatment, the adherent and floating cells were collected and washed with PBS and recovered by centrifugation at 300 rpm for 5 min. The cells were resuspended in PBS containing 7.5 µM of CM-H_2_ DCFDA and incubated for 30 min at 37 °C in a humidified CO_2_ incubator. The cells were washed twice with ice-cold PBS and analyzed on a Becton Dickinson FACScan flow cytometer (BD Biosciences Pharmingen, San Diego, CA, USA).

#### 3.7.2. Fluorescence Microscopic Evaluation of ROS

Cells were seeded in 6-well plates on sterile glass slides, and after 24 h they were treated with 15 µg/mL of holamine or funtumine for 12 h. After treatment, 2 µL of the NucBlue Live ReadyProbes^TM^ Reagent (Molecular Probes) was added to the growth medium and the cells were incubated for 20 min. After the removal of the medium, the cells were washed with HBSS. CM-H_2_DCFDA (2 µM) prepared in culture media (DMEM/F-12, no phenol red) was added directly onto the glass slides and the incubated at 37 °C for 30 min in the dark. After incubation, the cells were washed twice with HBSS and the slides were mounted on microscope slides. The resultant images were captured using a Zeiss Axio-Plan 2 fluorescence microscope (Zeiss, Germany).

### 3.8. Effect of Compounds on Mitochondrial Function

Mitochondrial function after exposure to the two compounds was analyzed by a cell-based (MitoToxGlo^™^) assay and fluorescence microscopy. 

#### 3.8.1. MitoToxGlo™ Assay

The cell membrane integrity and cellular ATP levels were assessed following the manufacturer’s instructions for the Topoisomerase-I relaxation assay kit (TopoGen, Buena Vista, CO, USA). The four cell lines were plated in a 96-well white walled plate (100 µL/well) at 5 × 10^4^ cells/mL in galactose-fortified glucose and FBS-free medium. The cells were treated with increasing concentrations (0–10 µg/mL) of the compounds for 2 h. The cells were first incubated with bis-AAF-R110 substrate for 30 min to measure the mitochondrial toxicity, and the fluorescence signal was recorded at 485 nm excitation and 585 nm emission wavelengths. Then, ATP detection reagent was added and further incubated for 5 min, after which the luminescence signal was measured.

#### 3.8.2. Fluorescence Microscopy

Cells were cultured in 6-well plates containing coverslips (22 mm × 22 mm) and treated with 15 µg/mL of the compounds for 12 h. The cells were stained with 5 µL of DAPI (4′,6-diamidino-2-phenylindole) and 250 nM of CMXRos for 30 min at 37 °C. The media containing the stains were aspirated, and the coverslips were washed in PBS and mounted onto microscope slides (25 mm × 75 mm) and visualized using a Zeiss Axio-Plan 2 fluorescence microscope.

### 3.9. Immunofluorescence Staining of Actin Filaments

The immunofluorescence staining of actin filaments in the cytoskeleton, focal contacts, and nuclei of the cells were evaluated using TRITC-conjugated phalloidin, anti-vinculin, and a DAPI staining kit according to the manufacturer’s instructions. Briefly, the cells were cultured on coverslips (22 mm × 22 mm) in a 6-well plate and treated for 3 and 6 h. The cells were fixed with 4% paraformaldehyde for 15 min, permeabilized with 0.1% Triton X-100 (in PBS) for 5 min, and blocked with 1% bovine serum albumin for 30 min. Anti-vinculin primary antibody (100 µL) was added and incubated for 1 h, followed by three washes with wash buffer. The resultant images were taken using a Zeiss Axio-Plan 2 fluorescence microscope.

### 3.10. Topoisomerase-I Relaxation Assay

This assay is based on the inhibition of the relaxation of supercoiled circular DNA by topoisomerase-I [51]. A master mix containing doubled-distilled water, 1X assay buffer (20 mM Tris-HCl (pH7.5), 200 mM of NaCl, 0.25 mM of EDTA, and 5% glycerol), 0.5 µg of supercoiled DNA, 1 U topoisomerase-I enzyme, and the test compounds were prepared according to the manufacturer’s instructions. The test compounds (15 µg/mL) were added to the Mastermix to a final volume of 30 µL. Preparation was conducted on ice before incubating the mixtures at 37 °C for 2 h. The reactions were terminated by adding 6 µL of 6X stop buffer (3% SDS, 60 mM of EDTA, 50% glycerol, 0.25% bromphenol blue). The reaction products were analyzed by 1% agarose gel electrophoresis. The gels were stained with 0.5 μg/mL of ethidium bromide for 30 min and visualized using a Biometra Image Analyzer (Biometra GmbH, Göttingen, Germany).

### 3.11. Statistical Analysis

The data generated were analyzed by two-way ANOVA followed by a post hoc Tukey’s multiple comparison test using GraphPad Prism version 8.0 for Windows (GraphPad Software, La Jolla, CA, USA) (www.graphpad.com).

## 4. Conclusions

The present study demonstrates that holamine and funtumine induced cytotoxicity through the induction of apoptosis in HeLa, MCF-7, and HT-29 cancer cells. The two steroidal alkaloids induced apoptosis through the elevation of ROS, mitochondrial function modulation, the perturbation of F-actin polymerization, and caspase-3 induction, which were all more prominent in HeLa cells. Moreover, the inhibition of DNA topoisomerase-I is an important function exhibited by the two alkaloids. The compounds show potential as anticancer agents that can play a role in translational medicine, however more mechanistic studies are still needed to unravel their full potential.

## Figures and Tables

**Figure 1 molecules-25-05716-f001:**
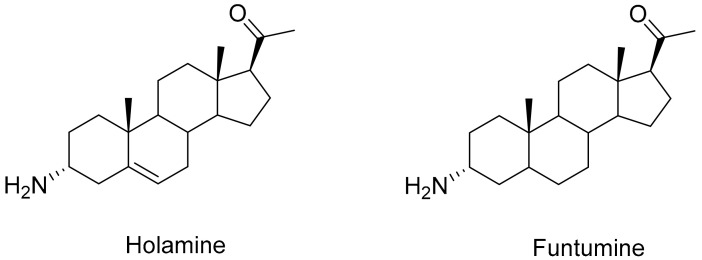
Structures of holamine and funtumine steroidal alkaloids isolated from *Holarrena floribunda*.

**Figure 2 molecules-25-05716-f002:**
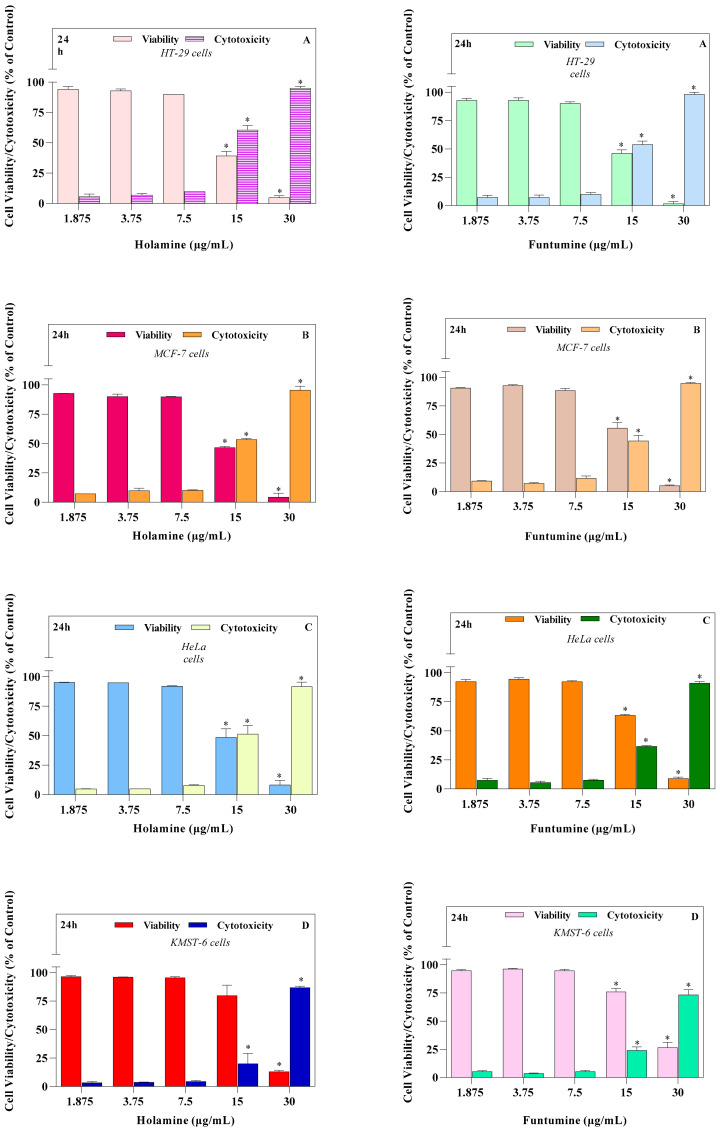
The cytotoxic effects of increasing concentrations of holamine and funtumine in (**A**) HT-29, (**B**) MCF-7, (**C**) HeLa, and (**D**) KMST-6 cells following a 24-h exposure, as evaluated by the CytoTox-Glo™ assay. Each bar represents the mean ± SEM of duplicate experiments. The * indicates statistical significance at *p* < 0.05 between the treatments analyzed—i.e., for viability assays in HT-29 cells, the effect elicited by 30 µg/mL of holamine differed significantly from those of the 1.875, 3.75, 7.5 and 15 µg/mL treatments. However, in the case of cytotoxicity, the effects elicited by 15 and 30 µg/mL of holamine differed significantly from those of the 1.875, 3.75 and 7.5 µg/mL treatments. The same statistical analogy holds for the other cell lines.

**Figure 3 molecules-25-05716-f003:**
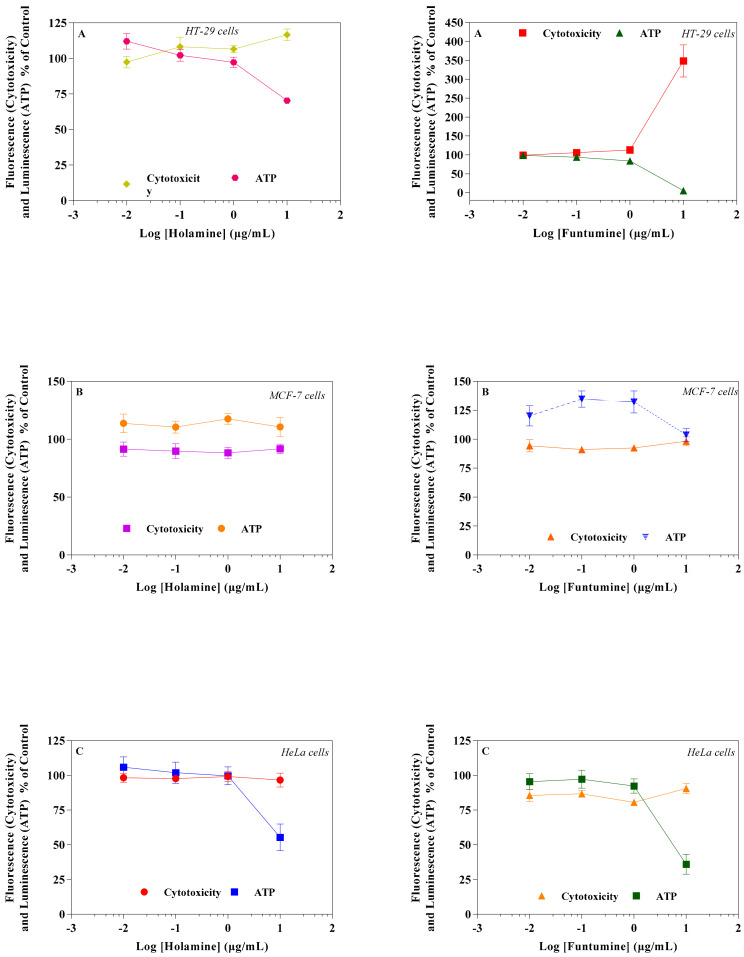
Effects of holamine and funtumine on mitochondrial function in HT-29I (**A**), MCF-7 (**B**), and HeLa (**C**) cancer cells. Cells were treated with increasing concentrations of the compounds for 2 h. Cytotoxicity and ATP production were measured by the MitoToxGlo™ assay. Values represent the means ± SEM of quadruplicate experiments.

**Figure 4 molecules-25-05716-f004:**
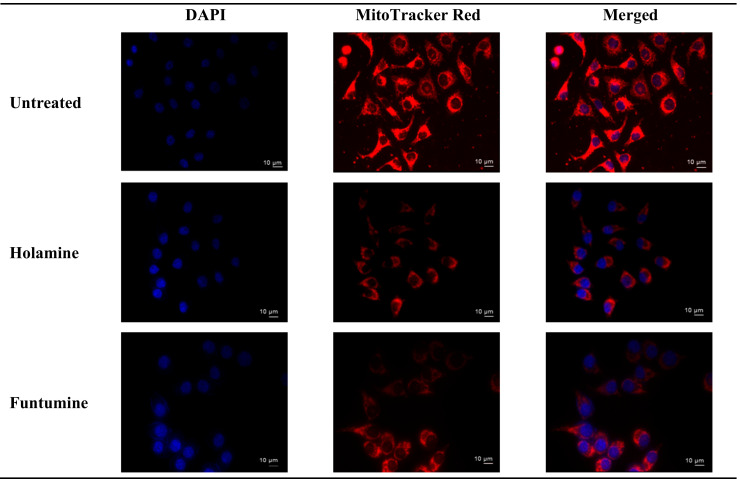
Effects of holamine and funtumine on HeLa cell mitochondria. Cells were stained both with Mitotracker red (red) and DAPI (4′,6-diamidino-2-phenylindole) (blue) after 12 h treatment and viewed by fluorescence microscopy (magnification × 400). Scale bars are indicated on the photomicrographs.

**Figure 5 molecules-25-05716-f005:**
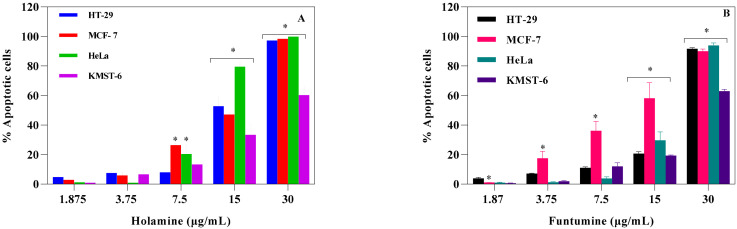
Apoptotic effects of holamine (**A**) and funtumine (**B**) on HT-29, MCF-7, HeLa and KMST-6 cell lines. Cells were stained with the APOPercentage™ dye and evaluated by flow cytometry. Each bar represents the mean ± SEM of triplicate experiments. The * indicates statistical significance at *p* < 0.05 among the cell lines for each concentration—i.e., for the apoptosis result in Panel **A**, the effect elicited by 7.5 µg/mL of holamine is significantly higher in MCF-7 and HeLa cells compared with HT-29 and KMST-6. The same statistical analogy holds for Panel **B**.

**Figure 6 molecules-25-05716-f006:**
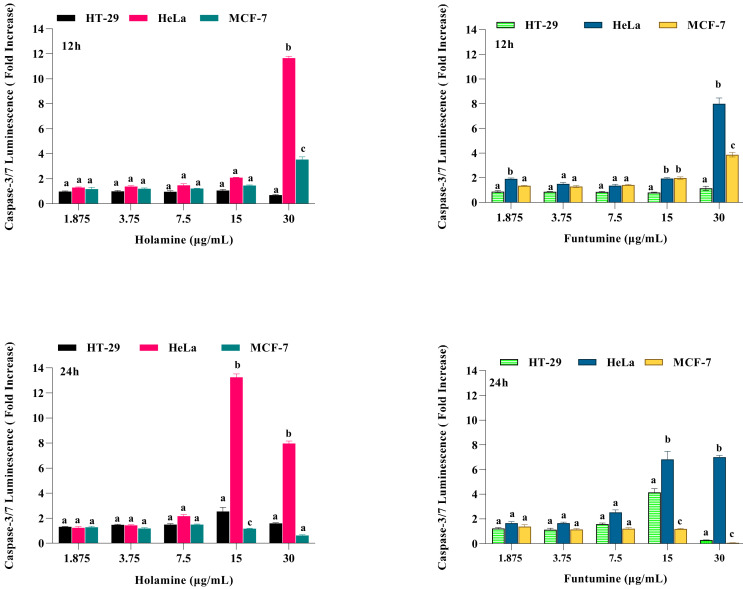
Effects of increasing concentrations of holamine and funtumine on the induction of caspase-3/7 activity in cancer cell lines (HT-29, HeLa and MCF-7) treated for 12 and 24 h, as indicated. Each bar represents the mean ± SEM of triplicate experiments. Bars that do not share a common alphabetic character differ significantly (*p* < 0.05).

**Figure 7 molecules-25-05716-f007:**
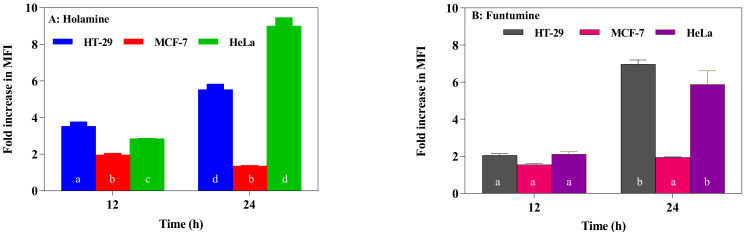
Reactive oxygen species (ROS) in the HT-29, MCF-7, and HeLa cancer cell lines treated with holamine (**A**) and funtumine (**B**). Cells were treated for 12- and 24-h periods with the compounds and thereafter stained with CM-H_2_DCFDA, and the results were evaluated using a flow cytometer. Data are presented as the mean fluorescence intensity (MFI). Each bar represents the mean ± SEM of triplicate experiments. Bars that do not share a common alphabetic character differ significantly (*p* < 0.05).

**Figure 8 molecules-25-05716-f008:**
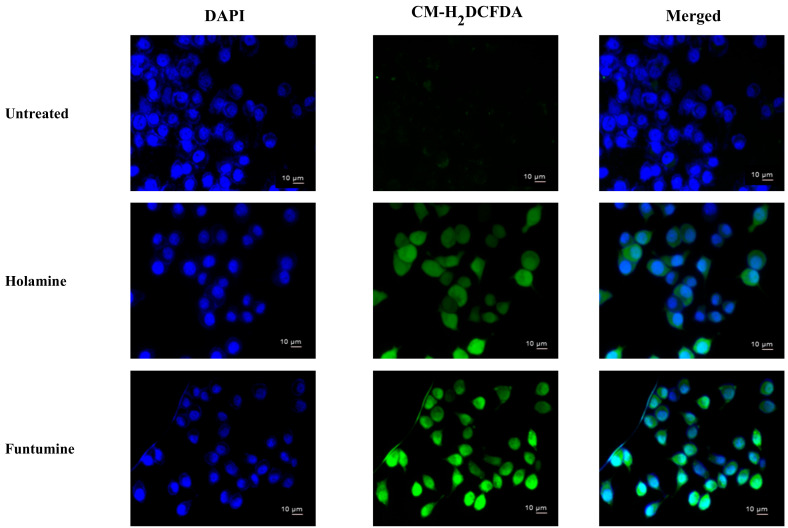
ROS induction after the exposure of HeLa cells to holamine or funtumine for 12 h. Cells were stained with 2 µM of CM-H_2_DCFDA and DAPI (4′,6-diamidino-2-phenylindole) and viewed by fluorescence microscopy (magnification ×400). Scale bars are indicated on the photomicrographs.

**Figure 9 molecules-25-05716-f009:**
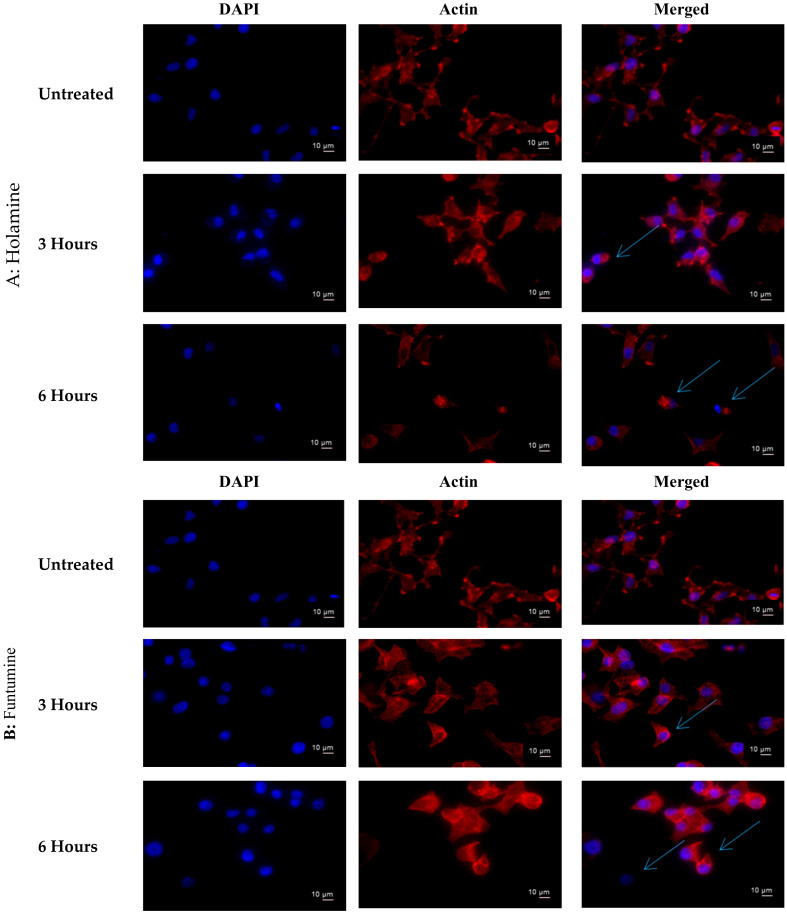
Effects of holamine (**A**) and funtumine (**B**) on the F-actin cytoskeleton of HeLa cells. Cells were treated with 15 µg/mL of funtumine for 3 h and 6 h, fixed and stained with Tetramethylrhodamine (TRITC)-conjugated phalloidin, and viewed under a fluorescence microscope (magnification ×400) to visualize the F-actin cytoskeleton. Blue arrows indicate the sliding over of the actin filament and the exposure of the cell nuclei. Scale bars are indicated on the photomicrographs.

**Figure 10 molecules-25-05716-f010:**
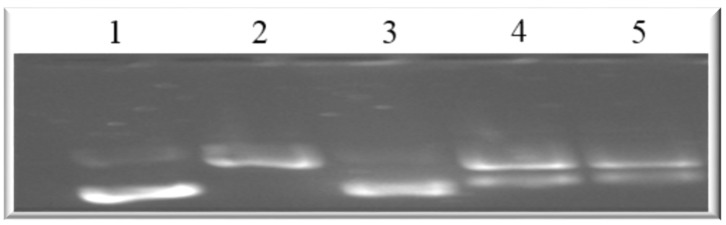
Agarose gel showing the DNA relaxation effects of holamine and funtumine on topoisomerase-I. Lane 1 represents supercoiled DNA; lanes 2, 3, 4 and 5 represent supercoiled DNA and with topoisomerase-I (2 U), camptothecin (9.3 mg/mL), and 15 µg/mL of holamine and funtumine, respectively.

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
