# Peer review of "An Insight into the Mechanism of Holamine- and Funtumine-Induced Cell Death in Cancer Cells"

_molecules, 2020, doi:10.3390/molecules25235716_

Round 1
Reviewer 1 Report
The article “An insight into the mechanism of holamine- and funtumine-induced cell death in cancer cells” reports the anti-proliferative mechanism of action of holamine and funtumine, two steroidal alkaloids isolated from Holarrhena floribunda (G. Don) leaves. The mitochondrial depolarization effects, reactive oxygen species (ROS) induction, apoptosis, F-actin perturbation and inhibition of topoisomerase-I was evaluated in three cancer cell lines (HT-29, MCF-7 and HeLa) and one non-cancerous cell (KMST-6). This manuscript needs minor revision before its publication.
Please provide more complete discussion about the holamine and funtumine effects on cancer cell lines (HT-29, MCF-7 and HeLa) and non-cancerous cell (KMST-6). The authors only refer “Figure 1 shows that the two compounds were significantly toxic to the cancer cells at the 15-30 µg/ml concentration range, and to the normal cells at the highest concentration (30 µg/ml).” but for example, the MCF-7 and HeLa cells present viability ≥ 50% at 15 µg/ml and other conclusions about the cells viability and cytotoxicity could be also discussed.
Page 7, line 121. The authors report “Furthermore, the compounds caused reduction in mitochondrial activity of cells after 6- and 12-h treatments as shown in Figure 3”. Figure 3 present the results after 6h treatment. No data is presented about 12 h treatment.
Page 8, figure 2, graph HeLa cells – Holamine. The standard deviations (SD) presented are high. The authors could provide values with better SD.
The fluorescence microscope images (Figure 3, 7 and 8) need to be improved. Scale bars are missing, the letters are overlaid by the images, font size is not the same on all images.
Page 11, figure 4. The authors could discuss better the results. For example: why at 15 µg/ml HeLa cells presents higher percentage of apoptosis when exposed to holamine and the other cell lines do not? or why at 15 µg/ml MCF-7 cells presents higher percentage of apoptosis when exposed to funtumine.
Suggestions:
- The authors could provide the chemical structure of holamine and funtumine.
- The authors could represent the information about cells evaluation with the same colour, could be facilitate the data analysis. See please figure 1 and 2.
(e.g. the viability graphic bar of holamine and funtumine on HT-29 cancer cell line could be represented with the same colour and the cytotoxicity graphic bar of holamine and funtumine on HT-29 cancer line the same -Figure 1).
Reviewer 2 Report
The manuscript from Badmus et al., describe the antitumoral effects of holamine and funtumine in terms of different parameters (e.g., cell viability, cytotoxicity, apoptosis activation, oxidative stress) in different cancer cells and, partially, compare it with the effects in non-cancerous cells. Although some interesting effects are presented, mechanisms are not general for all cancer cell types investigated, as claimed by the authors in several parts of the manuscript. Furthermore, some findings are highly contradicting. The following points need to be particularly addressed:
- The authors need to address the discrepancies between results presented in Fig 1 and Fig 2. Both Fig 1 and 2 present cytotoxicity data obtained with two different methods based on changes in the membrane integrity. While in Fig 1 an increase in cytotoxicity (i.e. loss of membrane integrity) was observed in HT-29 cells treated with holamine (15 ug/ml), MCF-7 and HeLa cells treated with holamine and funtumine (15 ug/ml), this is not the case in Fig 2 for the 10 ug/ml, where also a loss of membrane integrity is measured, although with a different method.
- Fig 2. Some panels do not show any error bars. Furthermore, statistical significance (as indicated in the text) also needs to be indicated in the figures.
- Fig 3. Images for HT-29 and MCF-7, now presented as data not shown, need to be included in the figure.
- Fig 3. A positive control affecting the mitochondrial membrane potential should be included in the figure.
- Fig 4. It is not clear which groups are compared in the statistic. For instance, in panel A, the % of apoptotic cells in MCF-7 cells treated with 15 ug/ml does not seem very different from KMST-6 cells. In panel B, the % of apoptotic cells treated with 15 ug/ml is almost the same for HT-29 and KMST-6 cells. Error bars in panel B are also missing. The statement in lines 167-170 needs to be consequently adapted since it applies only for the 30 ug/ml concentration.
- The conclusions from figure 5 also need to be adapted. Several conditions do not lead to an increase in caspase 3/7 activity.
- Figures 6 and 7, a compound inducing oxidative stress should be added as a positive control.
- Lines 69-71. The reference needs to be corrected. Reference 10 describes the antimicrobial activity of an extract of Holarrhena pubescens and not the cytotoxic effect on cancer cells.
- Section 3.8.1 and Fig 2. It is not clear how long the cells were exposed to the compounds. While the Materials and Methods section states 2 h, the figure legends states 24 h.
- Scale bars need to be added to all microscopy images.
Round 2
Reviewer 2 Report
The authors provided an improved version of their manuscript.
In the first revision, I indicated that the use of reference 10 to support antitumoral effects of Holarrhena pubescens extracts was not correct. The authors claimed that they have corrected it. However, they still indicate the same reference, which describes the antimicrobial effects of the above mentioned extracts. This must be checked and corrected.
Author Response
In the first revision, I indicated that the use of reference 10 to support antitumoral effects of Holarrhena pubescens extracts was not correct. The authors claimed that they have corrected it. However, they still indicate the same reference, which describes the antimicrobial effects of the above mentioned extracts. This must be checked and corrected.
Reply: We are sorry for this miss-up. The identify mistake has been corrected accordingly. Thanks